# Developmental Mapping of Hair Follicles in the Embryonic Stages of Cashmere Goats Using Proteomic and Metabolomic Construction

**DOI:** 10.3390/ani13193076

**Published:** 2023-09-30

**Authors:** Yuan Gao, Lei Duo, Xiaoshu Zhe, Lingyun Hao, Weiguo Song, Lizhong Gao, Jun Cai, Dongjun Liu

**Affiliations:** 1State Key Laboratory of Reproductive Regulation and Breeding of Grassland Livestock, School of Life Sciences, Inner Mongolia University, Hohhot 010070, China; 21908009@mail.imu.edu.cn (Y.G.); 21908010@mail.imu.edu.cn (L.D.); 22208021@mail.imu.edu.cn (X.Z.); imu_haolingyun@163.com (L.H.); 32008255@mail.imu.edu.cn (W.S.); 2Key Laboratory of Cashmere Materials and Engineering Technology in Inner Mongolia Autonomous Region, Ordos 010090, China; g_lz2004@126.com (L.G.); caijun1@chinaerdos.com (J.C.)

**Keywords:** cashmere goat, hair follicle, proteome, metabolome

## Abstract

**Simple Summary:**

The cashmere industry is a crucial resource and pillar of animal husbandry in the desert and semidesert regions of the central and western Inner Mongolia Autonomous Region. The development of embryonic hair follicles is vital in determining the quality and quantity of cashmere; however, the molecular mechanisms underlying embryonic hair follicle development remain unknown. This study employed proteomic and metabolomic techniques to analyse embryonic skin samples from cashmere goats. The results revealed that the oxytocin signalling pathway is crucial in embryonic hair follicle development by activating the MAPK and Ca^2+^ signalling pathways. The findings of this study expand our theoretical understanding of embryonic hair follicle development in cashmere goats and serve as a premise for future research.

**Abstract:**

The hair follicle (HF) is the fundamental unit for fleece and cashmere production in cashmere goats and is crucial in determining cashmere yield and quality. The mechanisms regulating HF development in cashmere goats during the embryonic period remain unclear. Growing evidence suggests that HF development involves complex developmental stages and critical events, and identifying the underlying factors can improve our understanding of HF development. In this study, samples were collected from embryonic day 75 (E75) to E125, the major HF developmental stages. The embryonic HFs of cashmere goats were subjected to proteomic and metabolomic analyses, which revealed dynamic changes in the key factors and signalling pathways controlling HF development at the protein and metabolic levels. Gene ontology and the Kyoto Encyclopaedia of Genes and Genomes were used to functionally annotate 1784 significantly differentially expressed proteins and 454 significantly differentially expressed metabolites enriched in different HF developmental stages. A joint analysis revealed that the oxytocin signalling pathway plays a sustained role in embryonic HF development by activating the MAPK and Ca^2+^ signalling pathways, and a related regulatory network map was constructed. This study provides a global perspective on the mechanism of HF development in cashmere goats and enriches our understanding of embryonic HF development.

## 1. Introduction

The cashmere goat industry is a crucial resource and pillar of animal husbandry in the desert and semidesert regions of the central and western Inner Mongolia Autonomous Region, playing a vital role in maintaining frontier stability, constructing ecological barriers, and promoting rural revitalisation [1,2]. Inner Mongolian cashmere goats are known for their fine cashmere, high strength, and high net cashmere rate. As the largest cashmere producer globally, China accounts for 70 percent of global production, with 30 percent from Inner Mongolian cashmere goats. Cashmere yield (the yield of raw cashmere) and quality (mainly based on fibre fineness and hand-tugging length) are two core indicators of Cashmere’s economic value. Cashmere of different quality has major differences in economic value and scope of application. However, the industry’s maintenance and protection of cashmere fineness traits are currently insufficient. Due to high demand in the international cashmere market, various producers have been performing crossbreeding and range expansion to increase production. This has led to a rapid decline in cashmere quality and the severe destruction of germplasm resources. Over the past decade, the average cashmere diameter (the average longitudinal projection width of cashmere fibres) in various regions of China has increased by about 1 μm.

Conversely, it is very difficult to stabilise cashmere traits due to factors such as population characteristics, rearing environments, and physiological diseases. The quality of cashmere is gradually declining. Although there are still a few populations of cashmere goats with a velvet fibre diameter of around 14.00 μm in the main production areas of Alashan, Inner Mongolia, and Ordos in the west, it is difficult to establish an industrial scale due to the small amount of ultra-fine-type cashmere goats.

The current traditional breeding methods are feasible to a certain extent; however, the insufficient diverse germplasm resources, long breeding time, and high breeding cost requirements make breeding high-quality-type cashmere goat populations extremely challenging. Genetic improvement programs for cashmere based on quantitative genetics also require the association of large amounts of genotypic and phenotypic data, and the process relies on phenotypic selection and empirical judgment, which does not accurately reflect the genetic level and potential of an individual or family line. Similarly, currently, molecular genetics is an important breeding tool. For example, Zhang et al. reported that melatonin promotes secondary hair follicle (HF) development in early postnatal cashmere goats and improves cashmere quantity and quality [3,4,5]. In addition, Liu et al. reported that *VEGF* and *Tβ4*-overexpressing cashmere goats exhibited faster HF development and a higher cashmere yield than wild-type cashmere goats [6,7,8]. However, considering the incomplete theory of HF biology in cashmere goats, especially on embryonic folliculogenesis, further research to comprehensively clarify the molecular mechanisms behind the phenotypic changes is limited.

In goat skin tissue, HFs are divided into primary HFs (PHFs) and secondary HFs (SHFs). PHFs produce coarse hair with a well-developed medulla; cashmere is the unmedullated fibre produced by SHFs. The development of HFs in cashmere goats primarily occurs during the embryonic stage [9,10]. The formation of PHFs begins around embryonic day 65 (E65) when all parts of the epidermis are completely formed and the basal, granular, and stratum corneum layers can be observed. Between E65 and E75, hair buds emerged and invaginated into the dermal layer; the PHF continued growing, and few SHF primordia were observed. Between E85 and E95, some PHFs were accompanied by SHF growth, representing the initial state of the ternary follicle group. By E105, the medulla of the PHF becomes apparent in the cross-section, and the SHF structure gradually forms. By E115, the PHF had penetrated the body surface; most PHFs had formed a complete structure, and some chorionic villi had penetrated the body between E125 and E135. Six months after birth, the development of the SHF structure is complete [11,12]. Based on studies and morphological observations, the critical stage of HF development occurs between E75 and E125 (Figure 1).

The HF is a crucial micro-organ in mammals, consisting of a relatively stable upper permanent region, including the funiculus and isthmus, and a lower follicular region, comprising the hair bulb and suprabulbar region [13]. The bulb is an essential component of the HF, responsible for developing and maintaining its morphology, and consists of internal inclusions and external tissues. HFs have a complex structure with various cell types, involving signalling pathway exchanges between different cell types and the spatial and temporal expression of numerous signalling pathways. For example, the activation of the Wnt/β-catenin signalling pathway promotes hair regeneration in mice [14]; exosomal miRNA-181a-5p from HF dermal cells promotes HF growth and development through the Wnt/β-catenin signalling pathway [15]. Using an HF induction model, Zhang et al. reported that the Wnt/β-catenin signalling pathway is required for NF-κB activation, with the direct target Wnt gene being the HF development gene *EDAR*; this indicates that these signalling pathways are interlinked in HF development [16,17,18].

Additionally, the pathway inducing HF atrophy can be inhibited by Dickkopf-associated protein 1 (DKK1): this phenomenon inhibits β-catenin phosphorylation, consequently inhibiting the Wnt signalling pathway [19]. In the Notch signalling pathway, when the receptor binds to a ligand, it activates HF stem cells, facilitating the transition of the HF from the resting phase to the anagen phase [20]. Signalling molecules in the Notch signalling pathway can synergise with those in the Wnt signalling pathway to perform biological functions during epidermal development. 

Furthermore, the BMP signalling pathway is involved in embryonic skin appendage morphogenesis and postnatal HF growth, with BMP2 and BMP4 genes inhibiting HF growth and development by blocking the entry of HF cells into the anagen phase [21,22]. In addition, noncoding RNAs (ncRNAs), especially miRNAs and lncRNAs, play a significant role in the regeneration and development of HF [23]. For instance, miR-184 affects SHF cell development via competitive targeting interactions with *FGF10*, while lnc-H19 functions in hair papilla cells through the chi-miR-19-214p/*β-catenin* axis [24,25].

Despite numerous studies on the molecular mechanisms of HF development, the global developmental mapping of HFs for the unique structure of cashmere goats is lacking. Notably, the mechanisms of action at the protein and small-molecule metabolic levels during HF development remain unclear, limiting the effective integration of various aspects of research and affecting the further development of molecular biology breeding in cashmere goats. Therefore, combining proteomic and metabolomic approaches is necessary to elucidate the molecular mechanisms underlying cashmere genes. 

This study systematically analysed the proteome and metabolome of skin samples from the embryonic stages (E75, E85, E105, and E125) of cashmere goats. Our data reveal the regulatory processes of proteins and metabolites in HF development, expounding the pathways in HF development and revealing unique differences between proteomes and metabolomes; additionally, a combined analysis of the proteome and metabolome provided relevant evidence for regulating HF development during the embryonic period. Constructing a global view of HF development during the embryonic period of the cashmere goat could facilitate a more comprehensive understanding of HF development in cashmere goats, especially SHF development.

## 2. Materials and Methods

### 2.1. Collection of Skin Tissues from Embryonic Chamois Goats

This study selected and grouped 24 female cashmere goats with clear pedigrees and similar production functions. After the oestrous cycle had commenced, artificial insemination was performed by using semen from the same ram and was recorded as embryonic development day 0 (E0). Pregnant ewes were then kept in a combined flock, with the temperature of the goat house maintained at 15–21 °C to ensure good ventilation and air quality. The grazing time was ≥12 h daily, with water consumed thrice. Each ewe was supplemented with at least 150 g of nutritious feed daily, including 1.5% salt, 1.0% trace elements, and 0.5% vitamins. When embryos reached E75, E85, E105, and E125, at least six goat foetuses were collected at each stage by using veterinary anaesthesia and inverted surgery. Samples were rapidly placed in liquid nitrogen for storage after collection, and the skin of cashmere goat foetuses was used for a subsequent histological analysis and morphological observation.

### 2.2. Haematoxylin–Eosin (H&E) Staining

After obtaining a goat foetus from the E75, E85, E105, and E125 stages, a 1 cm^2^ skin sample was obtained from the posterior side of the scapula on the somatic side of the foetus at the point where it crossed the midline of the abdomen and was used for morphological observations.

The skin samples were fixed with 4% paraformaldehyde for 24 h. The samples were then subjected to sequential ethanol gradient dehydration, xylene infiltration, paraffin embedding, and preparation as 7 μm sections by using a paraffin slicer. The sections were then sequentially stained by using haematoxylin, a differentiation solution, and eosin before being sealed with neutral resin and observed under a microscope.

### 2.3. Proteomic Analysis of Skin Tissues from Embryonic Cashmere Goats

Total protein was extracted from 12 skin samples (three from each E75, E85, E105, and E125 period). The BCA working solution and standard protein solutions of different mass concentrations were prepared by using a BCA kit (Thermo Fisher Scientific, San Jose, CA, USA), with mass concentrations ranging from 0 to 2.0 mg/mL. Each protein sample (2 µL) was diluted 10 fold with ultrapure water, and then 200 μL of BCA working solution was added. The reaction was carried out at 37 °C for 30 min, and the absorbance was 562 nm.

Subsequently, 100 μg of the protein sample was taken, and the volume was supplemented with lysate to 90 μL. Tris(2-carboxyethyl)phosphine (TCEP) reductant was added at a 10 mmol/L final concentration. The reaction was carried out at 37 °C for 30 min. Iodoacetamide was added at a final concentration of 40 mmol/L, followed by incubation for 40 min at room temperature while protected from light. TCEP reductant was added at a final concentration of 10 mmol/L, and the mixture was incubated for 60 min. The precipitate was separated by adding precooled acetone and fully solubilised with Tetraethylammonium bromide(TEAB) before digestion with trypsin at 37 °C overnight.

The Tandem Mass Tag (TMT) labelling reagent (Thermo Fisher Scientific) was removed from −20 °C storage and thawed to room temperature. Acetonitrile was added, and the mixture was vortexed. One tube of TMT reagent was added for every 100 µg of peptide. The mixtures were then incubated at room temperature for 2 h. Hydroxylamine was added, and the reaction was carried out at room temperature for 15 min. Afterwards, an equal amount of labelled product was combined in a tube, and the mixture was dried with a vacuum concentrator.

The peptide samples obtained were solubilised and separated on a reverse-phase C18 column for high-pH liquid-phase separation by using a UPLC buffer (2% acetonitrile in phase A, 80% acetonitrile in phase B, pH 10) at a UV detection wavelength of 214 nm with a volumetric flow rate of 200 μL/min and an elution time of 66 min. Based on their peak shapes and times, 20 fractions were collected from each group and combined into 10 fractions. They were concentrated by using vacuum centrifugation and dissolved with a mass spectrometry upsampling buffer for subsequent analysis.

A second-dimensional analysis was performed by using nanoscale liquid chromatography–tandem mass spectrometry (LC-MS/MS) with an EASY-nLC™ 1200 system (Thermo Fisher Scientific) and a Q Exactive mass spectrometer (Thermo Fisher Scientific). The peptides from the previous step were solubilised in the mass spectrometry buffer (phase A consisted of 2% acetonitrile, phase B consisted of 80% acetonitrile, all contained 0.1% formic acid) and separated on a C18 column for 120 min at a flow rate of 300 μL/min. Automatic switching between MS and MS/MS acquisition was performed with mass spectrometry resolutions of 70 K and 35 K, respectively. Full-scan MS was performed (m/z 350–1300), selecting the top 20 peptide parent ions for secondary fragmentation, with a dynamic elimination time of 18 s.

### 2.4. Metabolomic Analysis of Skin Tissues from Embryonic Cashmere Goats

According to the manufacturer’s instructions, 24 samples were prepared for subsequent experiments (six samples for each of the four periods of E75, E85, E105, and E125). For each sample, 50 mg of the skin samples was obtained, and 400 μL of the extraction solution (acetonitrile:methanol = 1:1) was added. The supernatant was pipetted for subsequent LC-MS analysis by using a Triple TOF5600 Triple Quadrupole Mass Spectrometer (AB SCIEX, Framingham, MA, USA). An equal volume of all the sample metabolites was mixed to prepare a quality control (QC) sample, with one QC sample inserted into every six samples before mass spectrometry detection. Mobile phase A consisted of water (containing 0.1% formic acid), and mobile phase B consisted of acetonitrile/isopropanol (1/1) (containing 0.1% formic acid). Next, mass spectrometry signal acquisition of the sample was performed in positive and negative ion scanning modes, with a mass scan range of *m*/*z*: 50–1000. The ion spray voltage was set at 5000 V for positive ions and 4000 V for negative ions, with a declustering potential of 80 V, spray gas pressure of 50 psi, an auxiliary heated gas pressure of 50 psi, curtain gas pressure of 30 psi, ion source heating temperature of 500 °C, and cyclic collision energy of 20–60 V.

### 2.5. Data Preprocessing and Database Annotation

Raw proteomic data were analysed by using ProteomeDiscovererTM Software 2.2. The search species was set as Capra_hircus, and the database used was Capra hircus (goat)|Proteomes|UniProt. During the process, the false discovery rate (FDR) for peptide identification was set to FDR ≤ 0.01, with proteins containing at least one specific peptide. The information of the identified proteins was compared with databases (COG, GO, KEGG, NR, Pfam, and String), with an annotation rate of 96.55–100%.

The raw metabolomic data were imported into the metabolomics processing software Progenesis QI (Waters Corporation, Milford, MA, USA) for analysis to obtain a data matrix. The data matrix was filtered, complemented, normalised, and log10 transformed to obtain the final data matrix for subsequent analysis. MS and MSMS information were annotated by using the public metabolic databases HMDB (https://hmdb.ca/, accessed on 25 March 2023) and Metlin (https://metlin.scripps.edu/, accessed on 25 March 2023) to obtain metabolite information.

### 2.6. Analysis of Differential Proteins and Metabolites

The preprocessed proteomic data were uploaded to the Majorbio Cloud platform (https://cloud.majorbio.com, accessed on 25 March 2023) for data analysis. The *t*-test function in R was used to calculate the *p*-value of significant differences between samples, and the fold change (FC) was calculated. The screening criteria for significantly differentially expressed proteins (DEPs) were as follows: upregulated proteins with *p* < 0.05 and FC > 2 and downregulated proteins with *p* < 0.05 and FC < 0.67. Afterwards, the differential proteins screened by the Student’s *t*-test method were subjected to an FDR correction, and each protein was arranged from small to large according to its *p*-value. Then, each protein multiplies its *p*-value by the total number and divides it by its number of arranged positions to obtain the FDR-corrected *p*-value. The calibration standard for this experiment is FDR ≤ 0.01.

For data analysis, preprocessed metabolomic data were uploaded to the Majorbio Cloud platform (https://cloud.majorbio.com, accessed on 25 March 2023). The R package ropls (Version 1.6.2) was used to perform a principal component analysis (PCA) and orthogonal partial least squares discriminant analysis (OPLS-DA), with seven cycles of cross-validation used to assess the model stability. Additionally, a Student’s *t*-test and multiplicative analysis of variance were performed. The selection of significantly different metabolites was determined based on the variable importance in projection (VIP) values obtained from the OPLS-DA model and the *p*-value of the Student’s *t*-test, with metabolites having VIP > 1 and *p* < 0.05 considered significantly different. The *p*-values were corrected for multiple testing by using FDR ≤ 0.01.

An analysis of variance was performed by using visualisation tools in Hiplot Pro (https://hiplot.com.cn/, accessed on 7 April 2023), a comprehensive web service for biomedical data analysis and visualisation.

### 2.7. Enrichment Analysis

Our study randomly divided data from four experimental periods into six control groups. We used proteins or metabolites with significantly different expression levels in each group for a subsequent functional enrichment analysis.

Gene ontology (GO, http://www.blast2go.com/b2ghome; http://geneontology.org/, accessed on 7 April 2023) was used for a functional clustering analysis of significant DEPs, and the Kyoto Encyclopaedia of Genes and Genomes (KEGG, http://www.genome.jp/kegg/, accessed on 7 April 2023) pathway database was used to analyse the metabolic pathways involved in the differentially expressed proteins.

Metabolic pathway annotation was performed by using the KEGG database (https://www.kegg.jp/kegg/pathway.html, accessed on 7 April 2023) to identify the pathways involved in significant differentially expressed metabolites (DEMs). The Python package scipy.stats was used for pathway enrichment analysis, and Fisher’s exact test was used to identify the biological pathways significantly relevant to the experimental treatments.

*p*-values < 0.05 represent significantly enriched biological functions and signalling pathways.

## 3. Results

### 3.1. Experimental Procedure and Morphological Identification of HFs

In this study, 24 skin samples were collected from embryonic cashmere goats, including six samples each from the E75, E85, E105, and E125 stages (Figure 2a). The obtained cashmere goat embryos were of the expected size. Skin sample collection was performed on the body side of the embryonic cashmere goats (Figure 2b). Morphological analysis of the embryonic skin samples from each stage revealed that during the E75 stage, the PHFs had developed a primitive morphology; however, the HF structure was incomplete, with the structure of the dermal papillae (DP) initially formed and SHFs not present. During the E85 stage, the PHFs had grown; further, the DP of the PHFs was complete, and SHFs began to occur, arising from the root of the PHFs and extending towards the skin surface. During the E105 stage, the PHFs and SHFs developed further. During the E125 stage, a trichome-type structure appeared (a follicular structure with three PHFs accompanied by multiple SHFs), with PHF development essentially complete, follicle morphology and structure fixed, and hairs and vellus protruding out of the body surface (Figure 2c).

These morphological analyses revealed that SHFs occurred from E75 to E85 and formed independent HFs from E85 to E105. Additionally, the basic unit of the cashmere goat HF (the trichome-type structure) was preliminarily formed from E105 to E125. The structural characteristics of samples within the same stage were uniform, with significant structural differences between samples from different stages. In summary, three skin samples from each stage were selected for proteomic analysis, and six skin samples were selected for metabolomic analysis in this study (Figure 2a).

### 3.2. Proteomic Analysis of HFs Derived from Embryonic Skin Tissues of Cashmere Goats

A proteomic analysis was conducted to investigate changes in protein expression during HF development in the embryonic period in cashmere goats. The PCA revealed good intragroup reproducibility and significant separation between samples from the four developmental stages (Figure 3a). A total of 273,662 secondary profiles were obtained; among them, matching with the protein database resulted in 73,912 profiles. Further, 44,401 peptides were identified, 13,381 proteins were identified, and 6384 proteins were functionally annotated for further analysis (Figure 3b). Six pairwise comparisons were performed between the four developmental stages (E75, E85, E105, and E125). Significantly differentially expressed proteins were identified by using statistical methods with a significance threshold of *p* < 0.05 and fold change cutoffs of 2 for upregulation and 0.67 for downregulation. Most DEPs were observed at the E125 stage. After removing duplicates, 1784 significant DEPs were identified (Figure 3c).

We summarised the significant DEPs obtained from six groups and removed duplicate DEPs. Through Z-Score standardisation, data of different levels were uniformly transformed into the same level for an expression-level clustering analysis. A cluster analysis of the expression patterns of the DEPs revealed five distinct clusters (Figure 3d). Cluster 1 contained 294 DEPs with a consistently elevated expression, Cluster 2 contained 427 DEPs with expression changes primarily concentrated in the E105–E125 period, Cluster 3 contained 198 DEPs with a consistently elevated expression from E75 to E125, Cluster 4 contained 330 DEPs with a consistently reduced expression, and Cluster 5 contained 535 DEPs with a significantly reduced expression from E85 to E125.

Additionally, the protein abundance of the top 10 DEPs was determined (Figure 3e), and all showed a higher abundance at E125 than at E75–E105; known DEPs, such as keratin KRT72, KRT71, and TET1, and unknown DEPs primarily belonging to the KRT family (Appendix A) were involved. These results suggest that the keratin family is highly active during the E125 stage and may play a crucial role in HF development.

### 3.3. GO and KEGG Analysis of Differentially Expressed Proteins

GO and KEGG analyses were performed for all the significant DEPs identified in this study. The GO enrichment analysis revealed that the top 20 GO terms were enriched in six pairwise comparisons between different HF developmental stages (Figure 4a). The results indicated that the biological functions of HF development primarily focused on developing the cytoskeleton and cellular structure. The most significant enrichment was observed for serine-type endopeptidase inhibitor activity (GO:0004867), and the highest number of proteins (121 DEPs) was enriched for supramolecular fibres (GO:0099512). Keratin filaments (GO:0045095), intermediate filaments (GO:0005882), and actomyosin structural organisation (GO:0031032) were significantly enriched in all six pairwise comparisons, suggesting their crucial roles in HF development.

The chronological grouping of HF developmental stages (E75 vs. E85, E85 vs. E105, and E105 vs. E125) revealed significant differences between the E75–E85 and E85–E125 periods, indicating two distinct stages of HF development with different biological functions (Appendix A). In the chronological grouping, myofibrils (GO:0030016), the myosin complex (GO:0016459), muscle contraction (GO:0006936), muscle system process (GO:0003012), sarcomere organisation (GO:0045214), and contractile fibres (GO:0043292) were significantly enriched, suggesting persistent roles in HF development (Figure 4a). The most significant enriched GO terms were observed in the E75 vs. E125 comparison, indicating a significant difference between the pre- and post-HF development periods (Appendix A).

In KEGG enrichment analysis, the top 20 KEGG pathways were identified and annotated for enrichment in the six pairwise comparisons between the HF developmental stages (Figure 4b). The results showed that the most significant enrichment was observed for the complement and coagulation cascade (chx04610) pathway, and the highest number of DEPs (297) was enriched for metabolic pathways (chx01100). In all pairwise comparisons, Staphylococcus aureus infection (chx05150) was significantly enriched, with significant keratin family members involved in its roles and changes. This suggests that keratins play an essential role throughout HF development and in skin keratinisation and skin derivative formation in later stages. Notably, the significant enrichment of DEPs was observed in the dilated cardiomyopathy (chx05414), hypertrophic cardiomyopathy (chx05410), and protein digestion and absorption (chx04974) pathways, indicating the crucial roles of DEPs at different levels of HF development through these pathways. In the chronological grouping of HF development stages, the cardiac muscle contraction (chx04260), adrenergic signalling in cardiomyocytes (chx04261), and vascular smooth muscle contraction (chx04270) pathways were significantly enriched (Appendix A).

### 3.4. Metabolomic Analysis of HFs Derived from Embryonic Skin Tissues of Cashmere Goats

This study conducted a metabolomic analysis to investigate changes in metabolite expression during HF development in the embryonic period in cashmere goats. A PCA revealed the good biological replication of samples from the four developmental stages and significant separation between groups, allowing for further data analysis (Figure 5a). Mass spectrometry was performed in positive and negative ion modes, yielding 3628 and 3769 mass spectral peaks, respectively; 830 metabolites were identified by using a search library; and 789 metabolites were annotated in public databases, such as HMDB and Lipidmaps. Four hundred ninety-four metabolites had annotated information in the KEGG database (Figure 5b). Six pairwise comparisons were performed among the four developmental stages (E75, E85, E105, and E125), and DEMs were identified by using a Student’s *t*-test. A total of 454 DEMs were identified (Figure 5c), with a significant increase observed during the E125 stage; combined with the protein data, this suggests that the E125 stage is a period of rapid HF differentiation, with a higher number of DEPs and DEMs compared to the previous developmental stages. Notably, most DEMs with significantly elevated expression levels in different pairwise comparisons were observed in proteomic data, indicating a strong correlation between proteins and metabolites.

A cluster analysis of all the significant DEMs was performed, and the top 50 metabolites were visualised in abundance (Figure 5d). The results showed that the DEMs were primarily clustered into five distinct expression patterns across the HF developmental stages from E75 to E125. Clustering heat maps and VIP (VIP represents the contribution value of the metabolite to the difference between the two groups, and the larger the value, the greater the difference between the two groups) bar charts were used to visualise changes in the importance and expression trends of the DEMs in all six pairwise comparisons (Appendix A). The top 20 abundant metabolites were analysed (Figure 5e). Although their expression was not significantly different, their high global expression suggests important roles (Appendix A).

### 3.5. Similarity Analysis and KEGG Analysis of Differentially Expressed Metabolites

To further understand the functional roles of the significant DEMs, a similarity analysis was performed on the top 50 DEMs in terms of metabolite abundance from the clustering analysis (Figure 6a). This allowed for the functional classification of known DEMs and the functional prediction of newly identified DEMs. The results showed that all the significant DEMs formed four modules with highly consistent correlations, suggesting that DEMs within the same module may have similar functional roles. A KEGG enrichment analysis was performed separately on DEMs from each module, and significantly enriched pathways (*p* < 0.05) were annotated. Module 3, with 28 enriched DEMs, showed high similarity and involved pathways such as apoptosis (map04210) and the sphingolipid signalling pathway (map04071). Two other modules, with 12 and 6 enriched DEMs, respectively, showed high similarity and were enriched in two signalling pathways. Finally, one module with three DEMs showed a consistent similarity, suggesting similar functional roles (Figure 6a).

A KEGG enrichment analysis was performed on all the significant DEMs, and the top 20 KEGG pathways were identified and annotated for their enrichment in six pairwise comparisons between HF developmental stages (Figure 6b and Appendix A). The results showed that the most significant enrichment was observed for nucleotide metabolism (map01232), and the highest number of DEMs (24) was enriched for the biosynthesis of cofactors (map01240). The significant enrichment of choline metabolism in cancer (map05231), cholesterol metabolism (map04979), and the sphingolipid signalling pathway (map04071) was observed in all six pairwise comparisons, suggesting that these are major signalling pathways for metabolite function during HF development. Notably, in the metabolite enrichment analysis, the E75–E85 and E85–E125 periods, which appear to be dependent on different metabolic pathways, differed in most of the signalling pathways in which they were enriched (Appendix A).

### 3.6. Joint Analysis of DEPs and DEMs

In this study, a VENN analysis was performed on KEGG signalling pathways that were significantly enriched (*p* < 0.05) in proteomic and metabolomic data (Figure 7a); 12 signalling pathways were significantly enriched in both data types (Appendix A). A joint KEGG analysis of DEPs and DEMs was performed based on the chronological grouping of the HF development stages (E75 vs. E85, E85 vs. E105, and E105 vs. E125), allowing for a better understanding of the dynamic changes and global distribution of KEGG signalling pathways during embryonic HF development (Figure 7b).

The results showed that all four pairwise comparisons were present in the oxytocin signalling pathway (chx04921 and map04921) and neuroactive ligand–receptor interaction (chx04080 and map04080). As an upstream pathway of the MAPK and Ca^2+^ signalling pathways, the oxytocin signalling pathway may play a sustained and crucial role in HF development. The neuroactive ligand–receptor interaction pathway plays extensive roles in organisms, involving neurotransmitters, hormones, and other molecules in transmembrane transport and receptor binding; its role in HF development is vital but not yet fully understood.

Furthermore, protein digestion and absorption (chx04974 and map04974), being significantly enriched in proteomic and metabolomic data (among the top 20 enriched pathways) and present in three pairwise comparisons in the joint KEGG analysis, showed a broad distribution of roles; this suggests an essential role of the protein digestion and absorption pathway during HF development. Additionally, several signalling pathways, such as the arachidonic acid metabolism pathway (chx00590 and map00590), were present in at least two pairwise comparisons, indicating that their roles warrant further investigation.

### 3.7. Network Mapping of the Action of the Oxytocin Signalling Pathway

This study constructed a functional pathway map for the oxytocin signalling pathway based on the results of combined proteomic and metabolomic analyses of embryonic HF development (Figure 8). The map illustrates the dynamic changes in DEPs and DEMs in this pathway. Based on existing research, it has been hypothesised that the hypothalamus acts as a signalling centre in which (oxytocin) OXT neurons release OXT, activating the MAPK and Ca^2+^ signalling pathways through the actions of OTR and Gq, consequently affecting HF development. These results suggest that the oxytocin signalling pathway (chx04921 and map04921), an upstream of the MAPK and Ca^2+^ pathways, may be involved in HF development by activating these pathways.

## 4. Discussion

Inner Mongolia white cashmere goats are selectively bred for their unique genetic characteristics and adaptability [26]. They are an essential resource for studying genetic improvement, biotechnological breeding, nutritional regulation, disease prevention and control, and the intensive management of cashmere goats.

The HF is the basic unit responsible for fleece and cashmere production in goats. Based on the existing research, this study provides further details on HF development during the embryonic period in cashmere goats. Morphological observations revealed that the period from E75 to E85 is characterised by the genesis of SHFs, with the most significant morphological feature being the emergence of SHFs from the roots of PHFs and their extension into the dermis, consistent with previous studies. From E85 to E105, Some SHFs form independent HF structures, separating from the root of PHFs and forming independent follicular structures associated with the epidermis. This phenomenon helps us understand the developmental process of SHFs during the embryonic period. From E105 to E125, a preliminary trichome-type structure was formed due to rapid follicle differentiation. Significant morphological changes occurred during this period, with PHFs essentially complete and primary hair shafts beginning to break through the body surface. Additionally, SHFs develop rapidly, forming a preliminary trichome-type structure along with the PHFs. In summary, these observations can enable a better understanding of the HF development process and further clarify the stages of HF development in cashmere goats.

The results of the present study provided insights into the expression of proteins and metabolites in the HF development stages of the embryonic period in velvet goats. Among them, the COL coding family proteins (e.g., COL5AS, COL14A, COL12A, and COL1A) were expressed at significantly higher levels in E75 than in the other three periods. The COL coding family proteins are the main components constituting the extracellular matrix (ECM), which enables the aggregation of the dermal papilla cells through their effects on cell adhesion [27], which is typically considered to have an important role in HF genesis. IGF2 can act through the IGF2/PI3K/AKT signalling pathway. Significant changes in expression during HF development may be related to its role in influencing cell differentiation and proliferation [28]. IGF2 may be involved in HF fibrogenesis in the velvet goat. The most significant changes in expression in the present study were observed in the keratin family (e.g., KRTAP11-1, KRTAP13-1, KRT35, KRT72, KRT71, etc.), and their expression was much higher in the E125 period than in the other three periods. Keratin families play important roles throughout HF development; however, during the rapid differentiation of HF from the E105 to E125 stages, significantly more keratins are involved in HF differentiation and are related to hair follicle morphogenesis and trichome development. In addition, keratin families are the most downstream group of target genes for HF development, and they have a direct role in hair follicle morphogenesis trait formation [29].

Concerning metabolites, prostaglandin H2 (PGH2) had high expression throughout HF development, and its expression was significantly higher during E125 than during the other periods. PGH2 is produced in large quantities downstream of the MAPK signalling pathway, converted to prostaglandin E2 (PGE2) and prostaglandin F2α(PGF2α), which alters intracellular signal transduction through specific binding to G-protein-coupled receptors (GPCRs), in turn altering cellular function. PGH2 may have an important role in the subsequent development of HF cells. Diacylglycerol (DAG) expression also changed significantly, and its expression was significantly higher in the period from E85 to E105 than in the E125 period. Some studies have shown that DAG is an important lipid second messenger, which can activate a variety of signalling cascades, such as protein kinase C (PKC) and Ras/Raf/MEK/ERK, and regulate various cellular functions, such as proliferation, differentiation, migration, and apoptosis [30].

The regulatory mechanisms of embryonic HF development in cashmere goats are not comprehensively understood. However, several signalling pathways and transcription factors play essential roles. In the present study, a KEGG enrichment analysis revealed that the most significant enrichment was observed in the complement and coagulation cascade (chx04610) pathway, which mainly plays a role in immune defence, coagulation, and the control of vascular permeability [31]. The pathway is associated with skin formation and angiogenesis in cashmere goat embryos at this period, which, in turn, influences HF development. In addition, due to the highly significant changes in the keratin family, there was significant enrichment of its associated *S. aureus* infection (chx05150) pathway. The pathway is usually associated with soft-tissue skin infections and pneumonia, among others, and its action is accompanied by drastic changes in the keratin family [32]. This is a side effect of the important role of the keratin family during HF development.

Interestingly, the significant enrichment of differentially expressed proteins (DEPs) was observed in the dilated cardiomyopathy (chx05414), hypertrophic cardiomyopathy (chx05410), and protein digestion and absorption (chx04974) pathways. The present study does not seem to be able to link them to HF development; however, all three signalling pathways have important roles in angiogenesis and altered permeability, as well as in inflammatory responses. We speculate that they may, in turn, influence HF development through their involvement in cutaneous angiogenesis and development. In contrast, producing different G-protein isoforms in adrenergic signalling (chx0426) alters cellular function by affecting downstream effector enzymes. This may be related to the strong cell proliferation and differentiation during embryonic life [33].

The present study showed that the oxytocin signalling pathway and neuroactive ligand–receptor interaction signalling pathway were enriched in pairwise comparisons between different developmental stages. The oxytocin signalling pathway typically plays a role in secretion, lactation, and regulating social behaviours and affective states such as prosociality and intimacy [34]. Oxytocin (OXT) produces the G-protein isoform Gq by binding to its G-protein-coupled receptor, oxytocin receptor (OTR), which can activate phosphodiesterases (PLCs) in effector enzymes to produce different second messengers, such as cyclic adenosine monophosphate (cAMP), diacylglycerol (DAG), and inositol triphosphate (IP3). These second messenger groups can activate the downstream Ca^2+^ signalling cascade response, affecting processes such as intracellular gene expression and protein translation, affecting cellular functions. The G-protein isoform Gq can also directly act on Ras and MEK5 in the MAPK signalling pathway to activate the MAPK signalling pathway. The MAPK signalling pathway regulates cell proliferation, differentiation, apoptosis, and other biological processes. It is closely related to Wnt/β-catenin, a key signalling pathway in HF development. The two pathways can regulate each other by directly or indirectly affecting the phosphorylation or degradation of β-catenin. In different cells and tissues, MAPK and Wnt/β-catenin signalling pathways can synergistically or antagonistically affect cell proliferation, differentiation, migration, and other functions. We speculate that during the embryonic HF developmental stage in velvet goats, the oxytocin signalling pathway may activate Ca^2+^ and MAPK signalling pathways. This, in turn, affects the downstream Wnt/β-catenin signalling pathway and ultimately impacts HF development.

The neuroactive ligand–receptor interaction pathway is a signal transduction pathway involving neuroactive ligands and receptors that can regulate cellular physiological functions such as neurotransmitter release, intracellular Ca^2+^ concentration, and gene expression. The link between this pathway and HF development remains unclear; however, it has been suggested to be associated with the presence of HF-associated pluripotent (HAP) stem cells. These cells may be the most primitive stem cells in the skin and can differentiate into neurons, glial cells, keratin-forming cells, and other cell types [35].

## 5. Conclusions

This study used proteomic and metabolomic techniques to construct a developmental atlas of HF development during the embryonic period in cashmere goats. A total of 6384 proteins were identified, with 1784 DEPs obtained from pairwise comparisons between developmental stages. Additionally, 789 metabolites were identified, with 454 DEMs obtained from different pairwise comparisons. GO and KEGG enrichment analyses were performed to provide comprehensive annotations of the developmental processes of HFs. A joint KEGG analysis revealed that the oxytocin signalling pathway plays a vital role in embryonic HF development by activating the MAPK and Ca^2+^ signalling pathways. This study provides a global view of the mechanisms underlying HF development in cashmere goats and enriches our understanding of embryonic HF development.

## Figures and Tables

**Figure 1 animals-13-03076-f001:**
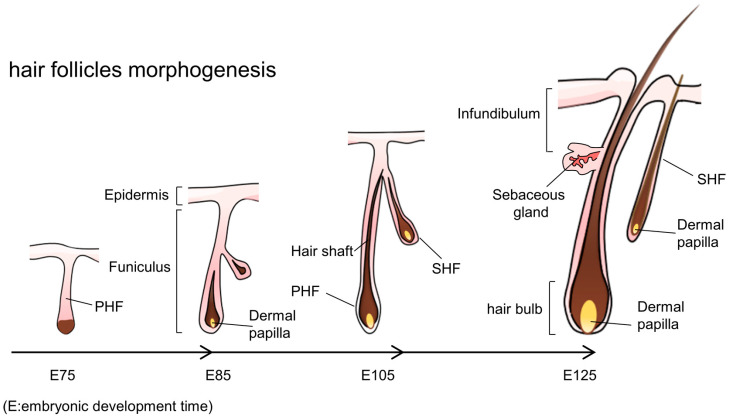
Hair follicle morphogenesis: E: embryonic development time; PHF: primary hair follicle; SHF: secondary hair follicle.

**Figure 2 animals-13-03076-f002:**
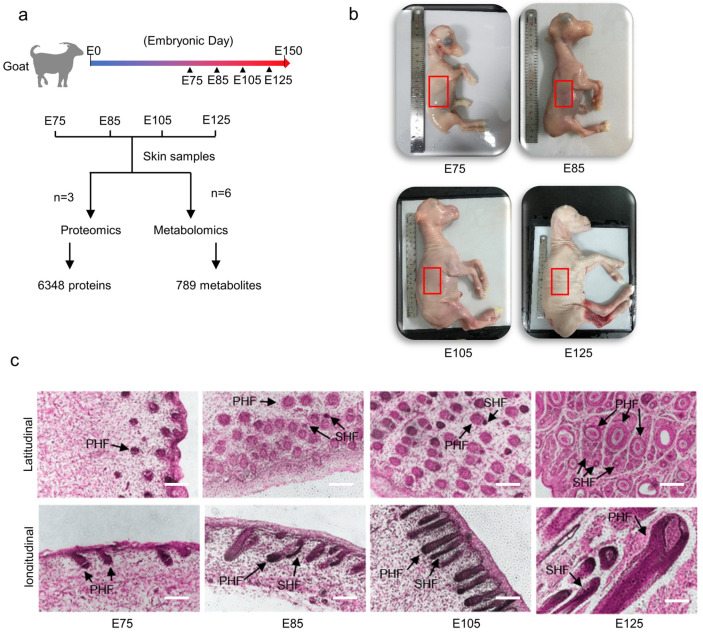
Experimental process and morphological identification of hair follicles (HFs): (**a**) Experimental process, with E representing the embryonic day. (**b**) Cashmere goat embryo, scale: 20 cm. The red area indicates where samples were collected. (**c**) Latitudinal and longitudinal sections of skin samples from E75, E85, E105, and E125 on a 20× scale (100 μm).

**Figure 3 animals-13-03076-f003:**
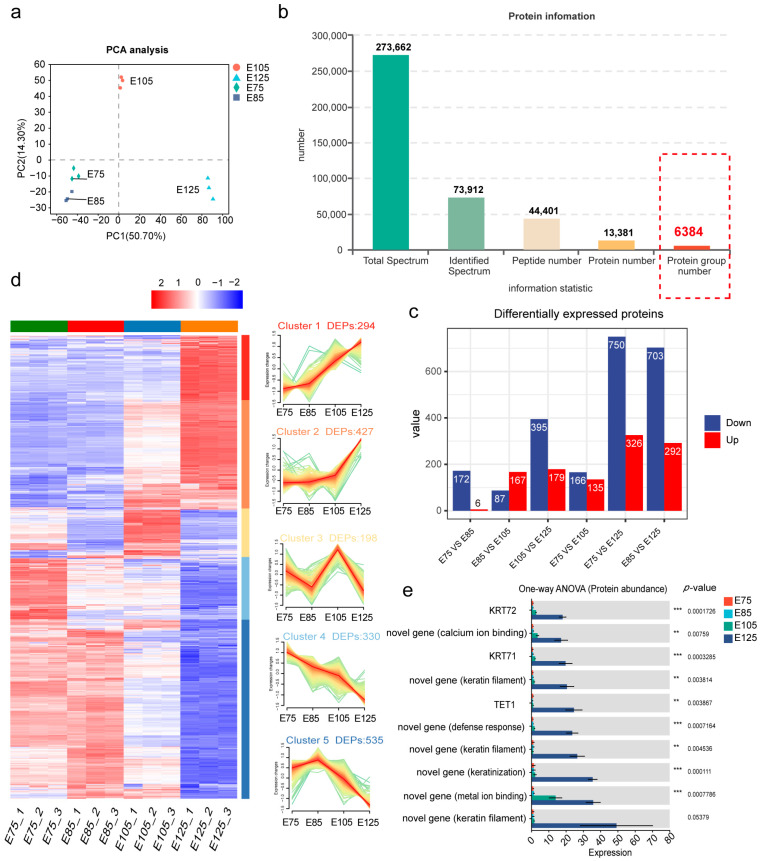
Results of the proteomic analysis: (**a**) principal component analysis (PCA) plot of the proteome, with different colours representing different samples and PC1 and PC2 representing the horizontal and vertical axes, respectively; (**b**) hierarchical filtration process for protein identification, with different colours representing different filtration levels; (**c**) counts of differentially expressed proteins (DEPs) in pairwise comparisons between different hair follicle (HF) developmental stages, with red indicating significant upregulated DEPs and blue indicating significant downregulated DEPs; (**d**) heatmap of expression clustering analysis, with samples on the horizontal axis and all significant DEPs on the vertical axis; (**e**) results of the univariate ANOVA analysis, including the top 10 proteins in terms of abundance, * (*p* < 0.05), ** (*p* < 0.01), and *** (*p* < 0.001).

**Figure 4 animals-13-03076-f004:**
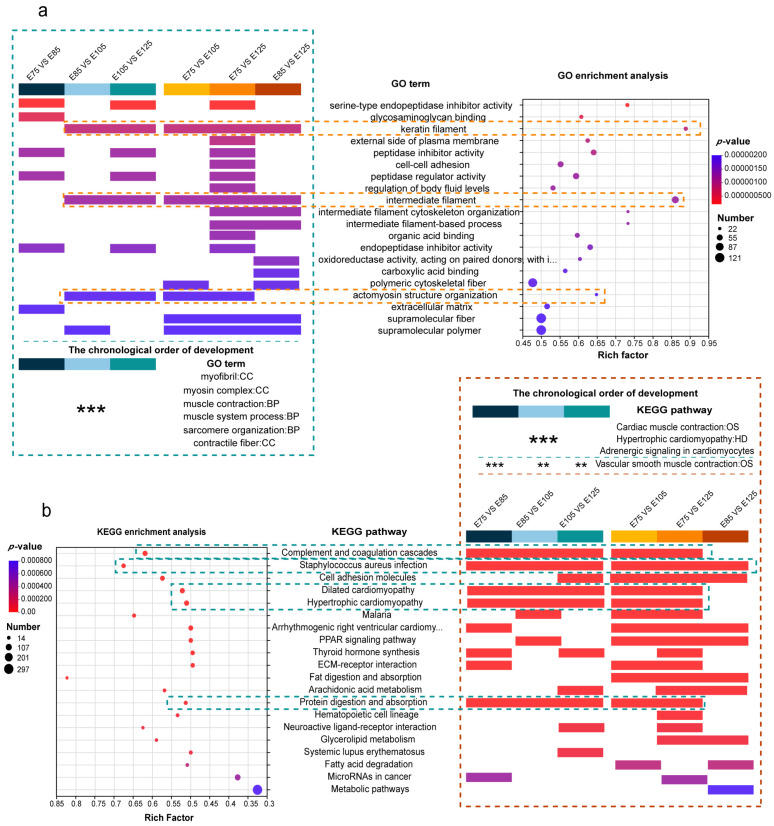
Results of gene ontology (GO) and Kyoto Encyclopaedia of Genes and Genomes (KEGG) analyses of the proteome: (**a**) GO enrichment analysis and annotation plots for different pairwise comparisons between hair follicle (HF) developmental stages. GO analysis was performed on all differentially expressed proteins (DEPs); the top 20 GO most significant terms were listed. GO terms were annotated if they were among the top 20 GO terms in the GO enrichment analysis for all six pairwise comparisons. GO terms that were significantly enriched in the chronological grouping of HF development stages were annotated, with significance levels indicated by * (*p* < 0.05), ** (*p* < 0.01), and *** (*p* < 0.001). (**b**) KEGG enrichment analysis and annotation plots for different pairwise comparisons. KEGG analysis was performed on all significant DEPs; the top 20 most significant KEGG pathways are listed. KEGG pathways were annotated if they were among the top 20 KEGG pathways in the KEGG enrichment analysis for all six pairwise comparisons. KEGG pathways that were significantly enriched in the chronological grouping of HF development stages were annotated, with significance levels indicated by * (*p* < 0.05), ** (*p* < 0.01), and *** (*p* < 0.001).

**Figure 5 animals-13-03076-f005:**
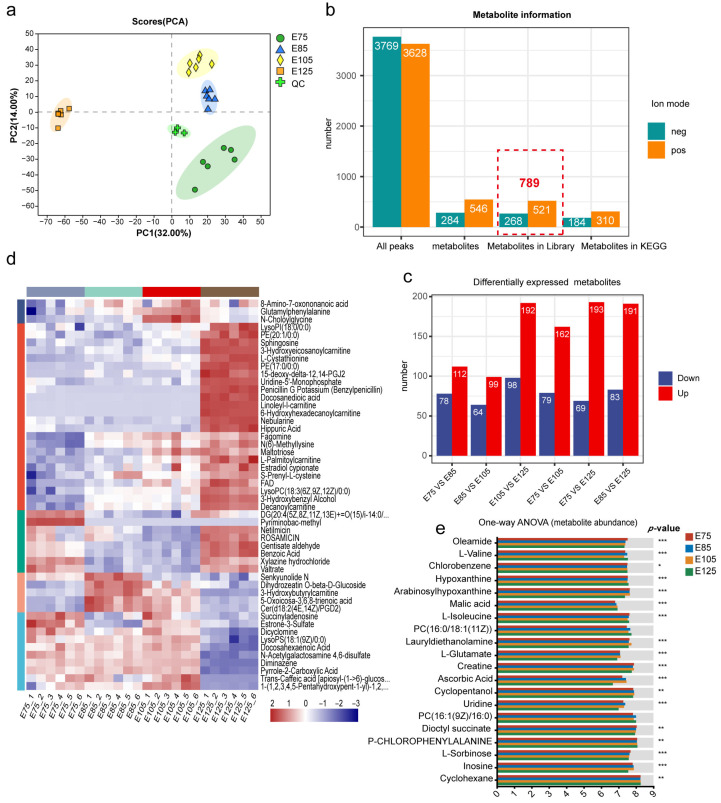
Results of the metabolomic analysis of embryonic hair follicle (HF) development. (**a**) Principal component analysis (PCA) plot of the metabolome, with different colours representing different samples and PC1 and PC2 representing the horizontal and vertical axes, respectively. (**b**) Layered filtration process for different ion modes in mass spectrometry, with positive and negative ion modes represented by “pos” and “neg”, respectively. Different levels of filtration are represented on the horizontal axis, and the number of metabolites is on the vertical axis. (**c**) Counts of differentially expressed metabolites (DEMs) in pairwise comparisons between different HF developmental stages, with red indicating significant upregulated DEMs and blue indicating significant downregulated DEMs. (**d**) Heatmap of expression clustering analysis, with samples on the horizontal axis and all significant DEMs on the vertical axis. (**e**) Results of the one-way ANOVA analysis, including the top 10 metabolites in terms of abundance * (*p* < 0.05), ** (*p* < 0.01), and *** (*p* < 0.001).

**Figure 6 animals-13-03076-f006:**
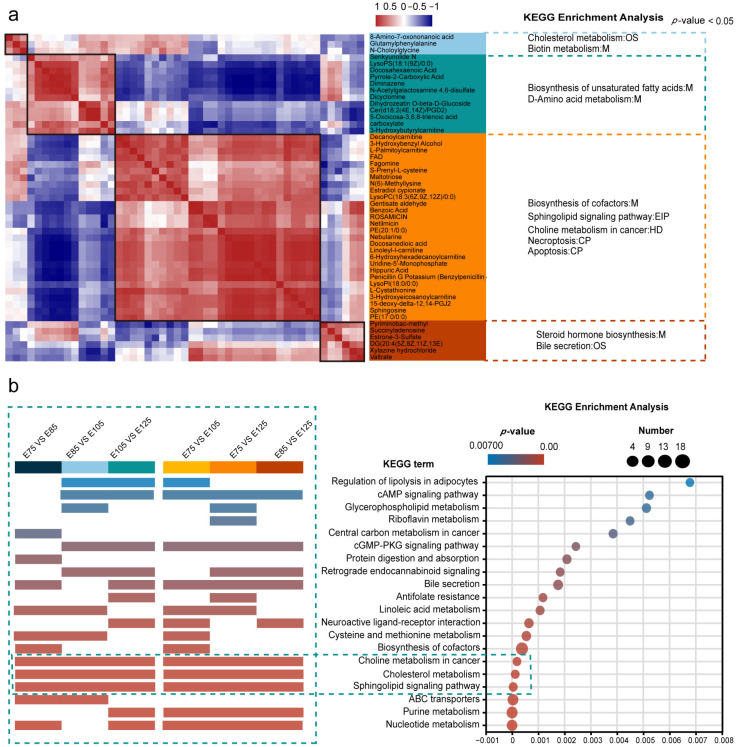
Results of the similarity analysis of differentially expressed metabolites (DEMs) and the Kyoto Encyclopaedia of Genes and Genomes (KEGG) analysis: (**a**) Similarity analysis of the top 50 DEMs in terms of metabolite abundance, with modules of different colours representing different similarity groupings. (**b**) KEGG enrichment analysis and annotation plots for pairwise comparisons between hair follicle (HF) developmental stages. KEGG analysis was performed on all significant DEMs; the top 20 most significant KEGG pathways are listed. KEGG pathways were annotated if they were among the top 20 KEGG pathways in the KEGG enrichment analysis for all six pairwise comparisons.

**Figure 7 animals-13-03076-f007:**
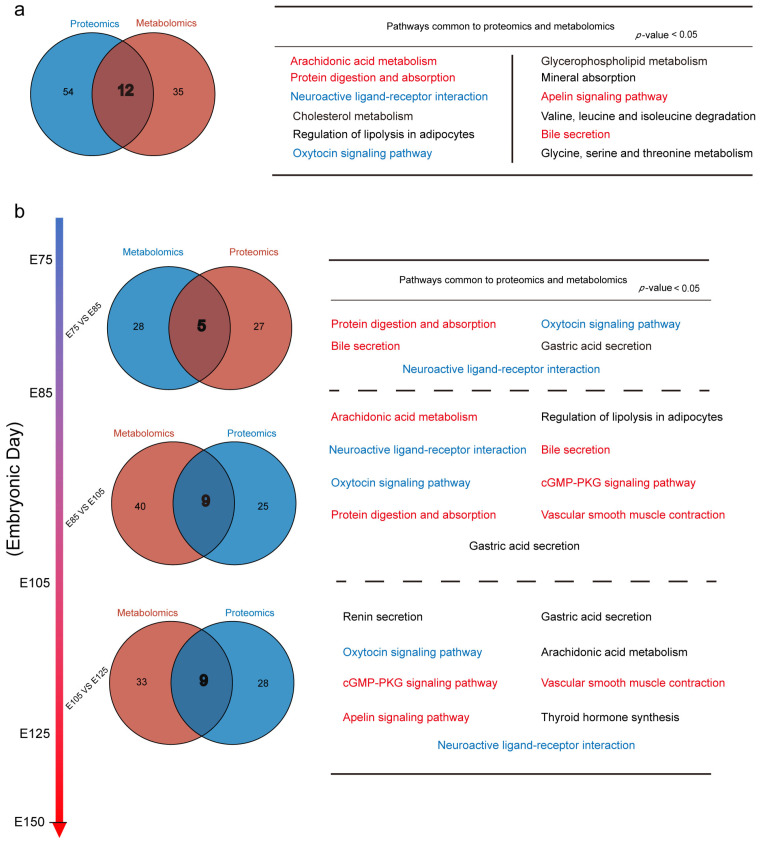
Results of the joint Kyoto Encyclopaedia of Genes and Genomes (KEGG) analysis of differentially expressed proteins (DEPs) and differentially expressed metabolites (DEMs): (**a**) VENN plot of KEGG pathways significantly enriched (*p* < 0.05) in proteomic and metabolomic data. (**b**) VENN plots of KEGG enrichment of DEPs and DEMs in pairwise comparisons based on the chronological grouping of hair follicle (HF) development stages, with pathways considered significantly enriched at *p* < 0.05. KEGG pathways in all four pairwise comparisons are marked in blue, and those in two or three comparisons are marked in red.

**Figure 8 animals-13-03076-f008:**
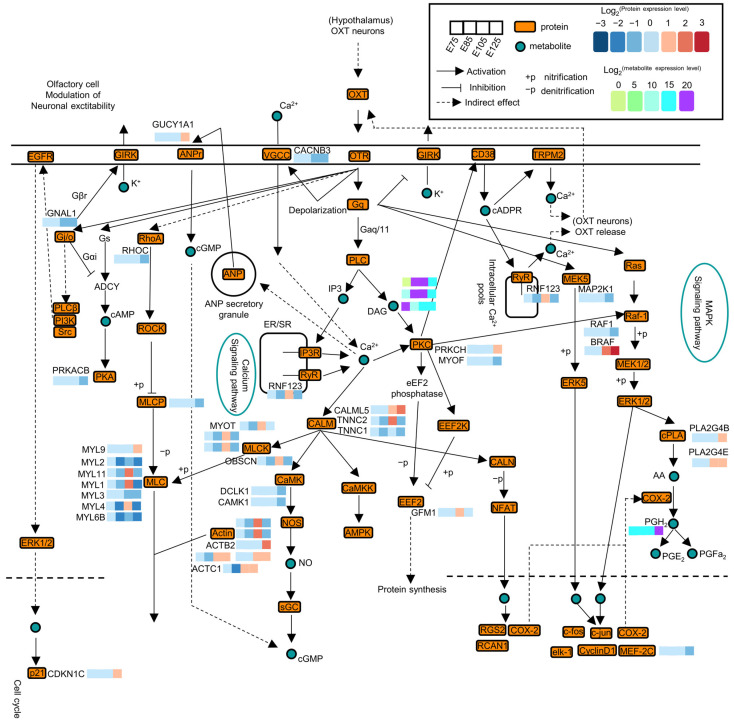
Schematic representation of the oxytocin signalling pathway, with yellow boxes representing proteins identified in this study and green circles representing metabolites identified in this study. Four consecutive boxes were used to represent expression at E75, E85, E105, and E125, with expression levels annotated by using log2 values relative to the expression in different colours. Solid arrows indicate direct action, dashed arrows indicate indirect action, solid flat lines indicate inhibition, “+p” represents nitration, and “−p” represents denitration. This visual representation can better explain the roles of the identified proteins and metabolites in the oxytocin signalling pathway during hair follicle (HF) development.

## Data Availability

All data are presented in the manuscript.

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
