# Peer review of "Developmental Mapping of Hair Follicles in the Embryonic Stages of Cashmere Goats Using Proteomic and Metabolomic Construction"

_animals, 2023, doi:10.3390/ani13193076_

Round 1

Reviewer 1 Report

The work remains interesting, given the importance and significance of cashmere production for the mentioned region. However, coincidentally, similar problems persist in this regard as in previous works of the same origin. The problem that you intend to solve with proteomics and metabolomics at the skin and hair follicle level is not clear. Is conventional genetic improvement feasible in these production situations? Or is there not enough knowledge to undertake a Cashmere genetic improvement program based on quantitative genetics?. Genomic selection or molecular genetics is currently an important complement to conventional Animal Breeding. Is there no possibility of applying it? Are there no facilities to implement a training population? etc. This should be clarified in order to justify such a sophisticated work as the one developed.

Furthermore, there are other problems that persist in this work, as in the previous ones reviewed by the same authors. There is a lack of proper usage of the terminology in Animal Science; common terms are used instead of specific terminology, making it difficult for potential interested parties, which include all specialists in the field. Coming from non-English speakers, we find it challenging to understand many cases. We recommend involving specialists in Animal Fiber Genetic Improvement to take part in the challenge of developing and writing these articles. For example, what do the authors mean by the word 'yield'? Is it referring to the total fiber production, the yield after dehairing, or the yield after washing?. And so on, bear in mind that the readers of these works can be geneticists, but also production experts who use a specialized language.

Only then will we be in a position to properly analyze the content of the work, with a clearly formulated hypothesis and appropriate terminology. 

Reviewer 2 Report

The authors investigated the cashmere hair follicle development in goats using proteomic data and metabolomic data. They did a huge work, however it is not clear whether the analysis was correctly done, because the methods do not describe exactly the starting material used for each sample and how it was done. For a description guideline, the following paper could be used:  https://doi.org/10.3389/fphar.2022.1040544. In addition, it is not clear whether the results were correctly analysed with correction for multiple testing.

In the discussion part, results of these papers should be compared to results of previous papers. For example, the authors do not put in perspective the KEGG enrichment results and it is not clear how the joint analysis find the oxytocin and Ca pathway to be upregulated.

Some minor comments

Introduction :

L58-96 : Figures could be added to facilitate the comprehension of the text with the key points of development of the HF.

L100-104: The role of these sentences is not clear: the function of the genes could be shortly named, or the authors want to focus on the used method, because the sentences do not bring information.

Methods:

L131: why at least 6, if only 24 females were used and 4 time points?

L137: samples of what? How were the samples prepared at the start.

L144: protein quantification: what is the starting material, how much material per sample, how was the samples homogenized and prepared?

L149: What is TCEP?

L150-152: At which temperature were the samples incubated?

L152: What is TEAB?

L153: What is TMT?

L155: How was the TMT-labeled peptides fraction obtained?

L156: what is the solvent?

L157-166: It is not clear how the fraction were prepared.

L168: What is used as the starting materials for the samples?

L191-3: Do the authors mean: “The data matrix was filtered, complemented, normalised, and log10 transformed  to obtain the final data matrix for subsequent analysis.

L198-211. 225: Were corrections for multiple testing applied?

Results

Fig 1 + L247: Why only 3 samples for proteomic? A schematic figure would help to understand the difference between the PHF and SHF or a figure accompanying the introduction.

L259: only half of the proteins are annotated, could another annotation be used?

L266: how were the clusters obtained?

L326: to discuss “Staphylococcus aureus infection »

L321-335 : discussion of this part ?

Discussion

L466-487 this part should be shorten

Some minor editing is required.

Round 2

Reviewer 2 Report

The authors improved the manuscript by adding welcome details, however the statistical part description is still weak and can be a source of untrusting these results.

I am not an expert on wollen production, but is the word fleece the right one for cashmere

L100- the different parts such as (funiculus and isthmus) could be showed on the Figure 1.

L177: The authors started with a maximum of 4ug of proteins (L174), how could they have 100 ug for proteins in L177, or L188 for the peptides?

L190: What is combined in a tube?

L234: The sample was normalized with what?

L241-5, L253-4: It is unclear how you correct for multiple testing in these sentences and why was the fold change limit below not the same as above (FC<0.67, FC>2.0).

L260: enrichment analysis, it is still unclear, how the analysis is done, details are missing.

L300: “matches to what?”

Figure 3: L322: are the counts normalized, how were they normalized, why not using a logistic regression to analyze the data?

          The clustering process is not described.

L328: “all DEPs identified in this study“. This means that not only the significative DE were taken for the analysis? If all were taken for the analysis, a correction for multiple testing should also be applied to this part.

L527-529: “The proteins are the main components constituting the extracellular matrix (ECM), which enables the aggregation of the dermal papilla cells through their effects on cell adhesion[27], which is typically considered to have an important role in HF genesis. The part “through their effects” is not clear.

L541: “PGH2” should be introduced or at least defined, as well “PGE2 and PGFa2.

The new sentences need to be checked.
